# Translation and Validation of the Individual Workload Perception Scale—Revised for Portuguese Nurses

**DOI:** 10.3390/healthcare10122476

**Published:** 2022-12-07

**Authors:** Catarina Cabrita, Pedro Lucas, Gisela Teixeira, Filomena Gaspar

**Affiliations:** 1Nursing Research, Innovation and Development Centre of Lisbon (CIDNUR), Nursing School of Lisbon; 1600-190 Lisbon, Portugal; 2Domus Aurea, Rua da Cruz, 2725-193 Sintra, Portugal

**Keywords:** management, nursing, work environment, workload, validation study

## Abstract

(1) Background: In Portugal, there is no scale to assess nurses’ satisfaction with their workload. Therefore, this study aimed to culturally/linguistically adapt and validate the IWPS-R scale, with 29 items, to produce a Portuguese version. (2) Methods: A quantitative, descriptive and cross-sectional study was conducted in two phases: translation and adaptation of the IWPS-R into Portuguese, following the steps recommended by Beaton; and evaluation of its psychometric properties in a non-probability sample of 323 nurses working in a hospital centre of reference in Portugal. (3) Results: The final scale obtained a total explained variance of 62.3% and a KMO of 0.88. The reliability was assured through the determination of an internal consistency of 0.88. The construct validity was determined through confirmatory factor analysis. The factorial model presented a good quality fit (X^2^/df = 2.118; CFI = 0.925; GFI = 0.900; RMSEA = 0.059; *p* (RMSEA ≤ 0.05) = 0.041; RMR = 0.078; MECVI = 1.525; SRMR = 0.0631) with five factors. (4) Conclusions: The Individual Workload Perception Scale-Revised-Portuguese version (IWPS-R-PT) is a 21-item tool with five dimensions: Nurse Manager Support, Team Support, Workload, Organizational Resources and Intent to Stay. The IWPS-R-PT was found suitable for hospital-based nurses and may be useful in other settings where nurses work.

## 1. Introduction

The World Health Organization (WHO) [1] recognizes the role that nursing plays in the quality of care provided, both in terms of number and presence of nurses in all health institutions and the responsibilities assumed within the organizational scope, which, in addition to providing care, include administrative–managerial and educational actions. On this point, the WHO [2] found that the approximately 28 million nurses currently working represent more than half of all health professionals in the world. Even so, there is a shortage of 5.9 million nurses worldwide, which gives a stressful profile to nursing work, with a consequently negative environment.

Nurses have professional expectations of career progression, articulated in responsibilities, rights and duties, working conditions and adequate remuneration. Given the current political, economic and social situation, the employment scenario has undergone changes in terms of employability in the nursing career, which is reflected in nurses’ satisfaction. Professional satisfaction is associated with better performance outcomes, which are essential to assessing the quality of services. Thus, it is strategic that nurses feel satisfied with their work [3].

One of the most impactful phenomena in social life is stress at work, and the International Labour Organization characterizes occupational stress as a threat to the health of workers [4]. Recognized as the main carers in the healthcare sector, nurses are central in organizations where they develop their roles, they are in the frontline in most of the healthcare systems and their contributions are essential to the development of goals and to the delivery of safe and quality care. This professional area is acknowledged to have high levels of stress associated with the work environment as nurses are constantly subjected to psychosocial risks with physical, psychological and social impact. This impact resulting from workload factors, interaction and conflicts with peers or other professionals, shift schedules, lack of material/technical/human resources and the emotional demands inherent in direct contact with patients [5,6,7,8]. The higher the workload, the greater the professional deterioration, and there is a cause–effect relationship that may compromise nurses’ professionalism [9] once they find it difficult to ensure safe and outstanding care, which causes stress [10].

Nurses’ working conditions have been the subject of growing concern and interest from researchers and managers, since these are drivers of the quality of care and of the professionals’ and patients’ levels of satisfaction [11]. Therefore, the nursing practice environment is highly influenced by professionals’ behaviours, productivity, motivation and satisfaction. For a favourable nursing work practice environment, it is necessary that all these aspects develop and thrive. Knowing the environments where nursing care delivery occurs is a strategy to contribute to improving it and, consequently, to improving the quality of nursing care [12,13]. The nursing practice environment is fundamental to the success of healthcare systems [14] and is related to organizations’ efficiency, quality of nursing care and patient safety [15,16,17].

A lack of support in and satisfaction with the work environment affects not only nurses but also patients, which promotes nurses’ intention to leave and hinders recruitment. They indicate that work overload and lack of organizational support are critical factors for their dissatisfaction and, consequently, their intention to leave [18].

Efforts to describe the complex question of nursing workload have focused on objective measures such as nursing hours per patient day and levels of patient dependency determined by the amount of care performed during a certain period. These methods do not consider the experience and the competencies of healthcare workers nor problems with nurse manager/peer support or resource limitations.

In the United States of America, there was no instrument available that could accurately and quickly assess nurses’ workload, taking into account the work environment when assessing their needs, or results that could provide strategies for institutional improvements and reverse nurses’ negative perceptions of their work environment. Therefore, Karen Cox developed the Individual Workload Perception Scale-Revised (IWPS-R), consisting of five dimensions: Manager Support, Peer support, Unit Support, Workload and Intent to Stay [18]. The Manager Support dimension is about how nurses understand their managers to be helpful and concerned about their needs. The Peer Support dimension concerns the relationship that nurses have with each other. The Unit Support dimension assesses the extent to which nurses feel they have access to materials, resources, supplies and services to carry out their work. The Workload dimension presents items that assess how pressure and overload dominate the nurses’ work environment. Finally, the Intent to Stay dimension seeks to measure the probability that nurses will remain in their job [18].

The selection of this instrument takes into account the suitability of the scale to assess nurses’ satisfaction regarding the workload that influences their work environment. In Portugal, there is no valid and reliable tool for nurses’ satisfaction regarding workload, and we therefore determined to translate and validate an existing tool, the IWPS-R, for Portuguese nurses. It has been widely used over the years in the United States of America, with strong indicators of validity and reliability [19,20,21,22]. It was also translated into and validated for two different languages, namely Mandarin [23] and Greek [24]. Both studies were conducted in central hospitals and presented very good reliability levels of 0.93 [23] and 0.88 [24], respectively.

The aim of this study was to culturally/linguistically adapt the IWPS-R and validate its psychometric properties to originate a reliable Portuguese version.

## 2. Materials and Methods

### 2.1. Study Design

The design of this study was quantitative, descriptive and cross-sectional, aiming to adapt and validate the Individual Workload Perception Scale-Revised.

### 2.2. Participants and Data Collection

The inclusion criteria of the study were nurses who work at the hospital and who agreed to complete the questionnaire. Nurse managers were excluded from our study.

The sample of this study was composed of nurses from a hospital centre of reference in Lisbon. After authorization from the Ethics Committee and complying with all ethical requirements, namely anonymity and confidentiality, the questionnaires were sent to the nurses’ institutional e-mails and were available for two weeks, from 10 February 2022 to 26 February 2022. The questionnaires included questions relating to sociodemographic, academic, professional and work characteristics. For validation, it is necessary to consider the size of the sample. For the translation and cultural adaptation of a measurement instrument, the minimum number of individuals to respond is calculated by a minimum sample of 5 individuals per item [25]. However, according to Gray et al. [26], for there to be some expression at a scientific level, it should be 10 individuals per item. Therefore, the minimum sample for our study was 290 participants. The sample was composed of 323 nurses. All those who answered the questionnaire completed all the items, therefore no questionnaire needed to be excluded.

### 2.3. Ethical Considerations

The scale’s author, Karen Cox, was contacted and gave authorization to translate and validate the original scale. The Ethics Committee and the Hospital Administration granted authorization to conduct this study (number 1152/2021). The respect of the participants’ autonomy and their right to confidentiality and anonymity was ensured, and all signed informed consent before their participation in the study.

### 2.4. Measurement Development

#### 2.4.1. Original Individual Workload Perception Scale-Revised

The Individual Workload Perception Scale-Revised was developed by Karen Cox in 2010 and derives from the original IWPS, which was initially created in 2002 with 46 items. The IWPS-R is composed of five subscales (Manager Support, Peer Support, Unit Support, Workload and Intent to Stay), with 29 Likert-type response items from 1 (strongly disagree) to 5 (strongly agree). The total variation of results may range from 29 to 145 points, and the higher the score, the higher the level of the nurse’s satisfaction with his/her workload [18,20,23]. The development of this instrument was based on Maslow’s Theory of Human Motivation, which exemplifies how an individual moves through a hierarchy of needs: physiological, safety, social, esteem and self-actualization. On this matter, when the first need is met (physiological), the person moves on to the next step and so on [18]. Thus, nurses must have their basic, lower-level needs met before managers can implement something for their higher needs. The IWPS-R instrument measures lower-level needs (physiological, safety and social). It is expected to take approximately 10 to 15 min to complete. The results’ scoring and interpretation process is also considered simple [18]. Concerning its validity, in Cox et al.’s original study [18], the internal consistency had a Cronbach’s alpha of 0.93 and the subscales ranged between 0.68 and 0.89.

#### 2.4.2. Translation and Cultural Adaptation—Content Validity

Following the methodology recommended by Beaton et al. [27], the scale’s translation from English into Portuguese was developed by two independent translators. One of the translators had nursing knowledge. We performed an analysis of both versions and generated a consensus version. A third translator then performed a back-translation of the consensus version. At this stage, we observed that all 29 statements back-translated into English preserved the same meaning as the initial statements. Lastly, a wide-ranging pre-test of the IWPS-R Portuguese version was piloted in a non-probability sample of nurses. In line with the literature [25,27], we decided that if more than 20% of these nurses had doubts about the scale, a reanalysis and a retranslation would have to be performed. The pre-test was applied in a sample of 25 nurses. The completion of the questionnaire took between 10 and 15 min, which is in line with the results of Cox et al. [18]. The 25 participants were asked to comment what they understood regarding the items’ meaning and whether they raised any doubts. In terms of understanding, 100% of the respondents answered that the scale was understandable. Ninety-two percent of the respondents had no problems completing the scale. The remaining 8% raised a relevant issue concerning the resemblance between three items (16, 20 and 22). This subject was clarified with the author, Karen Cox. The process of adaptation and validation was reviewed, and the difference between the items was discussed. It was concluded that the initial translations were faithful to the author’s explanations and, therefore, it was decided to keep them.

### 2.5. Factorial Analysis

To perform the exploratory factor analysis (EFA), the varimax method of rotation was used to extract the principal components. To analyse the adequacy of data to perform the EFA, two tests were used: the Kaiser–Meyer–Olkin (KMO) test and the Bartlett’s test of sphericity. KMO must be higher than 0.5, and both tests indicate the appropriateness of data for factorial analysis [27]. To perform the factor analysis, the items with factor loadings above 0.4 and with higher factor weights were retained [28]. The Kaiser criterion was chosen to extract the factors through the varimax orthogonal rotation technique. An analysis of the total variance explained by the results was developed. The Cronbach’s alpha coefficient (α) was the indicator selected to evaluate the instrument’s internal consistency and reliability. The values can range from 0 to 1, and a minimum of 0.70 much be achieved to ensure an acceptable reliability [28]. A confirmatory factor analysis (CFA) was sequentially performed to assess the quality of the model fit we obtained in the EFA. AMOS software (version 26.0, IBM Corporation, Armonk, NY, USA) was used to perform this statistical processing. The composite reliability was assessed as described by Marôco [28]. As recommended, different global adjustment indices were used, namely the ratio of chi-square to degrees of freedom (X^2^/df), comparative-of-fit index (CFI), goodness-of-fit index (GFI), root mean square error of approximation (RMSEA), root mean square residual (RMR), modified expected cross-validation index (MECVI) and standardized root mean square residual (SRMR). A good adjustment of the models is assumed when X^2^/df < 3 and the values of GFI ≥ 0.90 and CFI ≥ 0.95. Values of RMSEA, RMR and SRMR < 0.05 are considered ideal, although values between 0.08 and 0.10 are acceptable [28]. The quality of local adjustment was assessed by the factorial weights and by the items’ individual reliability. The model adjustment was based on the modification indices (greater than 11; *p* < 0.001). The statistical software used was IBM-SPSS Statistics version 27.0.

## 3. Results

Of a total of 323 nurses, the majority were women aged between 30 and 39 years. The results relating to gender agree with the reality of Portuguese nursing. Most of the nurses had a bachelor’s degree. On the other hand, 35.9% had post-graduate education, of which 15.5% had a nursing specialization course and 20.4% a master’s degree. The professional category of Nurse represented 73.7% of the sample. Regarding the type of unit, the majority of the sample works in inpatient wards. As for the time of professional activity, the majority of the sample had been in the profession for more than 15 years (50.5%). Likewise, concerning the length of professional experience in the organization, most reported longevity of more than 15 years (46.1%). On the other hand, with regard to the time of professional practice specifically in the unit where they worked, the sample was balanced between nurses with between 1 and 4 years (27.9%) and with more than 15 years (27.2%) in the current unit (Table 1).

### 3.1. Exploratory Factor Analysis

The KMO presented a value of 0.88 and the Bartlett’s test of sphericity 0.000 [29], which demonstrates good values for the principal component analysis.

In the EFA, items 1, 10, 11, 12, 13, 22, 23 and 25 did not meet the criterion of factor loading greater than 0.40, and so they were excluded. In the Portuguese version of the IWPS-R, 62.3% of the total variance was explained by five extracted components. This version is composed of 21 items in five dimensions: “Nurse Manager Support” (NMS) with seven items; “Team Support” (TS) with six items; “Organizational Resources” (OR) with three items; “Workload” (W) with 3 items; “Intent to Stay” (ITS) with two items (Table 2). We considered it crucial to keep the standpoint of Cox et al. [18]. Accordingly, we maintained the dimensions’ names, only adjusting the semantics and meaning of the dimension “Organizational Resources”.

### 3.2. Reliability Analysis

The validated IWPS-R presented a Cronbach’s alpha of 0.88, matching a level of internal consistency classified as very good [29], while the Cronbach’s alpha of the original scale was 0.93 [16]. The α fluctuated between dimensions (“Nurse Manager Support” α = 0.89; “Team Support” α = 0.87; “Organizational Resources” α = 0.83; “Workload” α = 0.80; “Intent to Stay” α = 0.78) (Table 2).

### 3.3. Confirmatory Factor Analysis

The five-factor model of the IWPS-R-PT fitted to a sample of 323 nurses revealed a poor quality of adjustment (X^2^/df = 2.550; CFI = 0.892; GFI = 0.869; RMSEA = 0.069; *p* (RMSEA ≤ 0.05) = 0.000; RMR = 0.082; MECVI = 1.764; SRMR = 0.0668). By suggestion of the modification index, seven trajectories were included in the model between the residuals of pairs of variables (B5–B9; B5–B24; B15–B29; B15–B26; B20–B21; B24–B27; B16–B26) who shared the same content, proceeding to the covariance of their errors. The final model showed a good quality of fit significantly higher than the original model in the study sample (X^2^/df = 2.118; CFI = 0.925; GFI = 0.900; RMSEA = 0.059; *p* (RMSEA ≤ 0.05) = 0.041; RMR = 0.078; MECVI = 1.525; SRMR = 0.0631).

Figure 1 illustrates the values of the five-factor model regarding the local adjustment, namely the standardized factorial weights and the items’ individual reliability. The items have standardized factor weights (λ) greater than 0.5, showing that all items have factor validity and individual reliability (λ^2^) greater than 0.25. There is evidence that the composite reliability is appropriate for all factors.

## 4. Discussion

Three items in the “Organizational Resources” dimension, relating to social services, chaplain, and psychological support, were excluded from the Portuguese version of the IWPS-R, just as they were excluded from the Taiwanese version [23]. According to Lin et al. [23], this may mean that these services have no influence on nurses’ satisfaction and the omission of these items has little impact on the nurses’ perception of their workload. The “Workload” dimension, in turn, was reduced to three items from the original scale, keeping the same items as the Taiwanese version. This indicates that in the Portuguese version there is no problem with double-load items between this dimension and the “Intent to Stay” dimension, as in the Taiwanese version. In this dimension, three items were excluded, which corroborates the comments submitted during the pre-test that mentioned the existence of three identical items (13, 17 and 23) about whether the nurse intends to leave his/her workplace in the next 12 months: 13—“I plan to stay in my current position for the next 12 months”; 17—“I plan to stay in my current position for at least the next 12 months”; and 23—“I intend to look for a new position in a different unit or in a different organization within the next 12 months”. Both items 13 and 23 were excluded, and only item 17 remained in the scale.

We also concluded that the “Team Support” and “Workload” dimensions demonstrate less influence on Portuguese nurses’ perceptions than on those of American nurses. On the other hand, “Organizational Resources” demonstrates less influence on nurses from Taiwan [23] as compared with nurses in Portugal, Greece and the USA [18,21]. As stated by Lin et al. [23], by analysing the conceptualisation of the dimension “Organizational Resources”, it is understandable that access to material resources and services is not as crucial as “Nurse Manager Support” in terms of satisfaction. In all validations of the scale [18,23,24,30], this is the dimension with the greatest weight in the nurses’ perception of their workload.

Based on a comparison of results of the factor structure between the scales, it can be concluded that, despite the small differences in items of the “Organizational Resources”, “Workload” and “Intent to Stay” dimensions, the content of items in the Portuguese version is conceptually compatible.

### Limitations

We consider that since the study was conducted during the COVID-19 pandemic, the answers may have been affected by work conditions. It would be important to extend this study to different settings of nursing practice at a national level, such as primary healthcare, long-term inpatient units and nursing homes.

## 5. Conclusions

This study was conducted to provide nurses, nurse managers and academia with a tool to assess nurses’ perceptions of their workload in order to contribute to improving favourable nursing practice environments.

The IWPS-R-PT with 21 items demonstrated an adequate factorial structure of five dimensions (Nurse Manager Support, Team Support, Organizational Resources, Workload and Intent to Stay). The Nurses’ Workload Perception Scale had good psychometric characteristics in the Portuguese cultural context. We highlight the Cronbach’s alpha of 0.88 of the validated scale, which indicates very good internal consistency, demonstrating a valid and reliable instrument to measure nurses’ satisfaction regarding their workload.

With implications for both research and practice, this study allowed us to verify the impact that the support of nurse managers, or those in management positions, and the nursing team has on a nurse’s individual perception of their workload. It is important to note the relevance of other validation studies of the same scale in different cultural contexts, such as in Taiwan and Greece, to compare with and support the results of the present study.

This study provides a pertinent data collection tool for nursing management and research in nursing settings where Portuguese is the major language to help understand and ultimately improve nurses’ perceptions of workload and thereby improve patient care. Therefore, it is our wish that this instrument will be useful and widely used both in scientific research and clinical practice.

## Figures and Tables

**Figure 1 healthcare-10-02476-f001:**
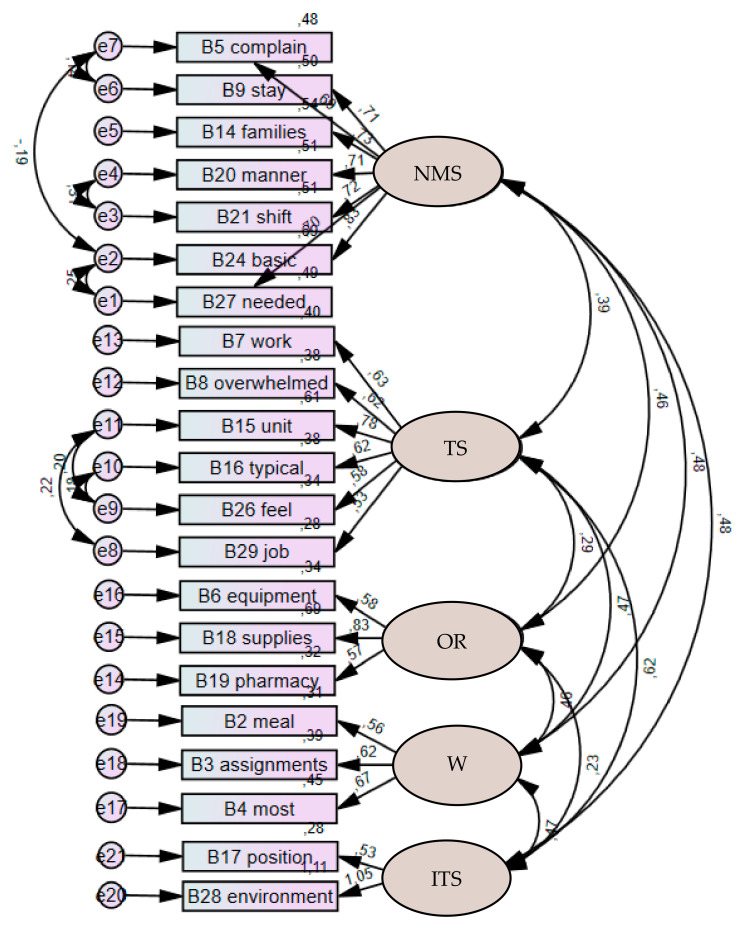
Five-factor model of IWPS-R-PT.

**Table 1 healthcare-10-02476-t001:** Sociodemographic, academic and professional characteristics.

Characteristics	Frequency (N)	Percentage (%)
Gender		
Female	282	87.3
Male	41	12.7
Age (years)		
20–29	89	27.6
30–39	101	31.3
40–49	71	22.0
50–59	59	18.3
>60	3	0.9
Level of nursing education		
Bachelor’s degree	207	64.1
Specialization	50	15.5
Master’s degree	66	20.4
Professional category	-	-
Nurse	238	73.7
Nurse specialist	85	26.3
Type of hospital unit		
Inpatient	133	41.2
Emergency Room	41	12.7
Intensive Care Unit	65	20.1
Operating Room	28	8.7
Outpatient clinic	21	6.5
Other	35	10.8
Length of professional activity (years)		
<1	17	5.3
1–4	49	15.2
5–9	44	13.6
10–14	50	15.5
>15	163	50.5
Length of professional activity in the organization (years)		
<1	21	6.5
1–4	68	21.1
5–9	53	16.4
10–14	32	9.9
>15	149	46.1
Length of professional activity in the unit (years)		
<1	33	10.2
1–4	90	27.9
5–9	67	20.7
10–14	45	13.9
>15	88	27.2

**Table 2 healthcare-10-02476-t002:** IWPS-R components.

ITEMS	Components				
	Nurse Manager Support	Team Support	Organizational Resources	Workload	Intent to Stay
1	0.63	-	-	-	-
2	0.72	-	-	-	-
3	0.80	-	-	-	-
4	0.72	-	-	-	-
5	0.69	-	-	-	-
6	0.85	-	-	-	-
7	0.80	-	-	-	-
8	-	0.68	-	-	-
9	-	0.62	-	-	-
10	-	0.78	-	-	-
11	-	0.71	-	-	-
12	-	0.64	-	-	-
13	-	0.67	-	-	-
14	-	-	0.72	-	-
15	-	-	0.78	-	-
16	-	-	0.66	-	-
17	-	-	-	0.74	-
18	-	-	-	0.75	-
19	-	-	-	0.58	-
20	-	-	-	-	0.88
21	-	-	-	-	0.62
Cronbach’s alpha	0.89	0.87	0.83	0.80	0.78

## Data Availability

Restrictions apply to the availability of these data. Data were obtained from a third party and are available with the permission of the third party.

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
