# Peer review of "Translation and Validation of the Individual Workload Perception Scale—Revised for Portuguese Nurses"

_healthcare, 2022, doi:10.3390/healthcare10122476_

Round 1

Reviewer 1 Report

See attached.

HEALTHCARE - 2052895 Brief Summary: This methodologic study examined the psychometric properties of the Individual Workload Perception Scale – Revised for Portuguese nurses (IWPS-R-PT). Several measures of validity and reliability were examined and found adequate. General Comments: This manuscript flowed logically and is well written with some correctible issues in throughout. Specific Comments: 1. Since the point of this study is nurses perception, retitle to “Translation and validation of the Individual Workload Perception Scale – Revised – Portuguese Translation (IWPS-R-PT)” This acronym should then be used throughout where appropriate, including the abstract to be consistent. 2. Abstract – Line 23-25: avoid using the word “proved.” Say instead, The IWPS-R-PT was found suitable for hospital based nurses and may be useful in other settings where nurses work. 3. Lines 69-73: Confusing as written, need to be more directly stated. Suggest “In Portugal there is no valid and reliable tool for nurses’ satisfaction regarding workload, we therefore determined to translate and validate an existing tool, the Individual Workload….developed by Karen Cox in 2006.” 4. Line 76: omit “a” before very good, and add an s to the word level so it reads “…presented very good reliability levels of…” 5. Line 86 “the Lisbon region”… 6. Lie 90: Do you know your return rate? Is the 323 nurses representative of the population of nurses who work in hospitals in Portugal? This is important data to know if you want to generalize to all of hospitalized nurses in your country and certain is of interest to other countries so all can compare your typical nurse population 7. Line 108: “It was performed an analysis of…” makes no sense. Did you mean the research authors did this or the translators did the analysis of the two translations? I think you mean that the two translators met and generated a consensus version what was then back translated by a third translator. Please clarify. 8. Line 119-120: “in terms of understanding,….” This sentence should follow the sentence I lines 121-120 so it is clear the 8% refers to the 92% in line 118. 9. Line 139: you say “as per described in” should read “as described by” 10. Line 152: you say “to request the ..” should say “ was contacted and gave authorization..” so it is clear you not only asked but she gave you permission to use. I would also ask if the journal or you or Dr. Cox maintains the copyright of this tool 11. Lines 159-172: this entire paragraph needs major rewriting. Please do not repeat so much was can be viewed in the table. Instead, summarize such as “Most respondents were women, aged 31-39, had a nursing Diploma, held a professional nurse license, ….. “ Then if you want to say something about what was unusual (such as a most were very experienced) say that. I would also like you to state if this sample represents the population of nurses in the same settings as you used here. 12. Table 1: a. Relabel “Title 1” as “Characteristics”, then for each of those Bold the Characteristic such as “Gender” not Sex by the way and Left align it. This will set off the characteristic from the options. b. You need to add the age in also. If you have the mean and standard deviation for age use it, but if done as categories, then list them. c. Relabel “Academic Qualifications” as “Education” or “Preparation” and start with lowest to highest degree only. This is more internationally recognizable. Since you list “Specialization” separately without any explanation it is difficult to know what this means. I would put it at the very end of this list and then asterisk with a note as what this means. In the USA this would be a specialty certificate on top of the degree earned. d. You need to address what you mean by Professional Category for the international reader: do you mean a Professional Nurse educated at the Diploma or Bachelor’s level versus the Nurse Specialist who has a Master’s or higher degree? Please asterisk and explain. Is the Professional Category a license designation for practice? e. Type of Hospital unit: 16.8% (outpatient clinic and other) of your respondents are not hospital-based nurses. This is a significant number of your respondents. Given that clinics are nothing like inpatient in terms of workload, and who knows what you have in the other category, this could skew your results. Please address in in your discussion section. 13. Line 197: IWPS-R-PT This is the first you use this acronym, please spell out. 14. Line 199-200: confusing. Do you mean to say “By suggestion of the modification index, seven trajectories were included in the model between…” 15. Line 207: Omit the phrase As it can be observed. And just start the sentence with “As items have…. 16. Figure 11 page 7: WAY too small to read. Needs to be enlarged or the numbers inlarged. Also add a list of the abbreviations below it. 17. Line 220 omit “the” item and just say only item 17.. 18. Line 246: omit “Regarding the limitations of the study”, since that is the title of this section. Start with “We consider that since the study was conducted during….” 19. Line 260: change “proving to be “ to “demonstrating a valid and …” You can never really Prove anything with human research, only demonstrate associations. 20. Line 266-267 Omit from “once it allowed…” to end of the sentence as this phrase is redundant. 21. Lines 268-271: need major rewrite. Suggest something like this: “This study provides a pertinent data collection tool for nursing management and research in nursing settings where Portuguese is the major language to help understand and ultimately improve nurses perceptions of workload and thereby improve patient care.”

Reviewer 2 Report

Dear authors, firstly congratulate you on the work you developed. The article presents an investigative coherence in its development. However, minor details mentioned in the comments need to be improved.

Title: okay.

Summary: okay.

MeSH: Not all keywords are MeSH. Find on page.

Introduction: In defining concepts and problematization, the main actors in this subject, such as the WHO and the International Labor Organization, should be included.

Methodology:
Study type:
Participants: the inclusion and exclusion criteria, and the number of choices of the sample size of both the translation and validation experts, need to be declared.
Data Collection:
Data analysis: okay
Ethical aspects: okay.

Results: okay.

Talk: okay.

Conclusions: okay.

References: include the relevant actors mentioned previously.

Reviewer 3 Report

The current manuscript presents the validation of the Individual Workload Perception Scale – Revised (IWPS_R) an instrument used to measure the nurses experience and perception of nurses regarding the workload, in the Portuguese healthcare context. A cross sectional design was used and data were collected from a sample of 323 nurses in Portugal. The authors present factor analysis of the IWPS_R demonstrating a good internal validity of the instrument in the Portuguese language. However, they did not use other instrument that measure similar constructs to those of the target scale in order to concurrently assess the external validity of the instrument, which represents the greatest limitation of the present study.  

Introduction

The authors do not provide a clear and satisfactory description of the IWPS_R, which is the main focus of this work and the instrument that they are trying to validate. Being this a validation study, the introduction section should focus more on a detailed description of this measure and its components (the subscales) while grounding its importance in the literature that demonstrates how important to nurse wellbeing are factors such as team support, management support or organizational resources. Authors should provide more details about subscales of the measure with examples of items which can render the idea more clearly and the cited literature should focus on more on the dimensions measured through this scale.

Methods

The “Ethics” section should be placed prior to that of the translation of the instrument.

Figure 1 needs to be reviewed as some of coefficients cannot be seen clearly due to overlaps with figure features.

Discussion

In lines 214-218 authors state “Three items of the dimension “Organizational Resources”, related to social services, chaplain and psychological support, were excluded from the Portuguese version of the IWPS-R, just as they were excluded from the Taiwanese version [20]. According to Lin et al. [20], this may mean that these services have no influence on nurses' satisfaction and the omission of these items has little impact on the nurses' perception of their workload”.

It is not clear why the authors decided to exclude these items? Making reference to another validation study in a totally different context does not offer an appropriate explanation. Further more, one might expect that what this may mean in the Taiwanese context may have little relevance for the Portuguese context. Authors should elaborate on this point.

In the same lines the following passage in lines 230-234 is quite confusing “We also concluded that the “Team Support” and “Workload” dimensions demonstrate less influence on the Portuguese nurses' perceptions than on the American nurses. On the other hand, the “Organizational Resources” demonstrates less influence in the study with nurses from Taiwan [20], comparing with the Portuguese nurses in this study and the nurses in the studies from Greece [21] and USA [15].”

Round 2

Reviewer 3 Report

No further comments.